# Beyond Natural Images: A Dual-Stream DINOv3 Framework for PET/CT Segmentation

**Yu-Nong Scarlett Lin**[1,2]                   YLIN828@GATECH.EDU
**Shansong Wang**[2]                   SHANSONG.WANG@EMORY.EDU
**Mojtaba Safari**[2]                   MOJTABA.SAFARI@EMORY.EDU
**Xiaofeng Yang**[1,2,*]                   YXFBME@GMAIL.COM

[1] *Department of Biomedical Engineering, Georgia Institute of Technology and Emory University, Atlanta, GA, USA*

[2] *Department of Radiation Oncology and Winship Cancer Institute, Emory University, Atlanta, GA, USA*

[*] *Corresponding author*

**Editors:** Accepted for publication at MIDL 2026

## Abstract

Self-supervised vision transformers like DINOv3 are strong universal feature extractors, yet their transferability to functional medical imaging remains limited when pretrained on misaligned natural-image domains. In this work, we introduce Dual-DINOv3, a dual-stream framework for PET/CT that addresses two key gaps in existing work: the absence of a public, PET-specific pretrained encoder and the reliance on fully paired PET/CT data for multimodal pretraining. First, we presented the first PET-specific DINOv3 encoder, pretrained exclusively on large-scale public FDG-PET datasets using the full three-stage DINOv3 self-distillation pipeline. Second, we proposed a modality-separated PET/CT framework in which PET- and CT-specific encoders are pretrained independently and fused during finetuning via multiscale cross-attention, enabling multimodal representation learning without requiring paired data during pretraining.

Evaluation on the HECKTOR tumor segmentation benchmark demonstrated three central findings: (1) misaligned natural-image pretraining degrades PET/CT performance relative to training from scratch, (2) domain-aligned CT pretraining substantially improves segmentation across all tumor sizes, and (3) dual-stream PET/CT pretraining achieves the best performance overall, highlighting the complementary contributions of functional and anatomical cues. Together, these results provide a fully public PET encoder and a scalable PET/CT foundation model that support domain-aligned representation learning under realistic clinical data constraints.

**Keywords:** Self-supervised learning, DINO, PET/CT, Foundation Model, Segmentation

## 1. Introduction

Self-supervised vision transformers such as DINOv3 (Siméoni et al., 2025) have emerged as powerful universal dense feature extractors capable of supporting diverse downstream tasks, including classification, segmentation (Li et al., 2025; Yang et al., 2025), and registration (Wang et al., 2025). Their self-distillation pipeline removes the need for labeled data, substantially reducing the barrier to domain adaptation in medical imaging where annotations are costly and inconsistent. However, models pretrained solely on natural images transfer

poorly to medical modalities whose physics and appearance diverge from RGB signals–a limitation evident in whole-slide pathology, electron microscopy, and especially positron emission tomography (PET) (Liu et al., 2025). MedDINOv3 (Li et al., 2025) partially addressed this limitation by introducing a computed tomography (CT)-pretrained DINOv3, demonstrating that aligning pretraining with the target imaging modality markedly improves downstream segmentation. These findings reinforce a central principle of medical foundation models which is domain alignment matters.

PET/CT plays a central role in diagnostic imaging and radiation oncology (Gafita et al., 2025), a role further enhanced by recent advances in long–axial field-of-view total-body scanners that enable high-sensitivity quantification of metabolism, inflammation, hypoxia, and metastatic spread (Dimitrakopoulou-Strauss et al., 2023; Cook et al., 2025). As clinical PET applications diversify across anatomical sites and radiotracers, there is a growing need for universal segmentation models capable of handling heterogeneous targets and acquisition protocols. Yet progress on PET- or PET/CT-specific foundation models has been limited primarily due to the scarcity of large, publicly available datasets.

Existing PET foundation efforts face key constraints. SegAnyPET (Zhang et al., 2025), trained on the private PETS-5K dataset (5,731 volumes; 1.3M slices), shows strong generalization using a segment-anything-model-like promptable architecture (Ma et al., 2024), but reliance on private data restricts reproducibility. In the multimodal domain, Oh *et al.* (Oh et al., 2025) proposed a PET/CT foundation model trained on paired inputs to learn cross-modality correspondences; while effective, this design requires paired PET/CT scans for all pretraining data, limiting scalability in real-world settings where PET-only or CT-only datasets are much more common.

As a result, to our knowledge, there is no fully public and reproducible PET-specific foundation encoder pretrained at scale on public PET datasets, and existing PET/CT foundation efforts either rely on private data or require paired PET/CT scans during pretraining. At the same time, large-scale public paired PET/CT datasets suitable for foundation-model pretraining remain limited, making it challenging to develop reproducible multimodal PET/CT encoders using only public resources. Motivated by this constraint, we propose a modality-separated strategy that pretrains PET and CT encoders independently on large public single-modality datasets and learns multimodal fusion only during downstream finetuning when paired PET/CT data are available. This work makes the following contributions:

- **A Public PET-Specific DINOv3 Encoder.** We develop the first PET-specific DINOv3 encoder pretrained entirely on publicly available PET datasets using the full three-stage DINOv3 pipeline, providing a reproducible and domain-aligned representation model for functional imaging. The pretraining datasets involved multi-institute, multi-dose level fluorodeoxyglucose (FDG)-PET imaging covering a wide range of axial field-of-view from brain to pelvic 2D slices.

- **A modality-separated PET/CT framework.** We introduce a dual-stream PET/CT architecture in which PET and CT encoders are pretrained independently on unpaired public datasets and fused only during downstream finetuning, avoiding reliance on large-scale paired PET/CT data during pretraining. Standard cross-attention is used as a fusion mechanism to integrate modality-specific representations.

## 2. Materials and Methods

This section details the full Dual-DINOv3 training pipeline, consisting of (i) domain-specific pretraining of independent CT-DINOv3 and PET-DINOv3 encoders, and (ii) domain-fusion finetuning for PET/CT tumor segmentation (Figure 1). We first describe the construction and preprocessing of the large-scale CT and PET database used for single-modality DINOv3 pretraining, followed by the downstream PET/CT datasets used for supervised segmentation. We then outline the three-stage DINOv3 self-distillation objective, the implementation of modality-separated pretraining, and the cross-attention feature fusion mechanism used to combine PET and CT representations during finetuning. Finally, we present the model configurations, cross-validation protocol, and evaluation metrics used to quantitatively assess the benefit of modality-specific pretraining and multimodal fusion.

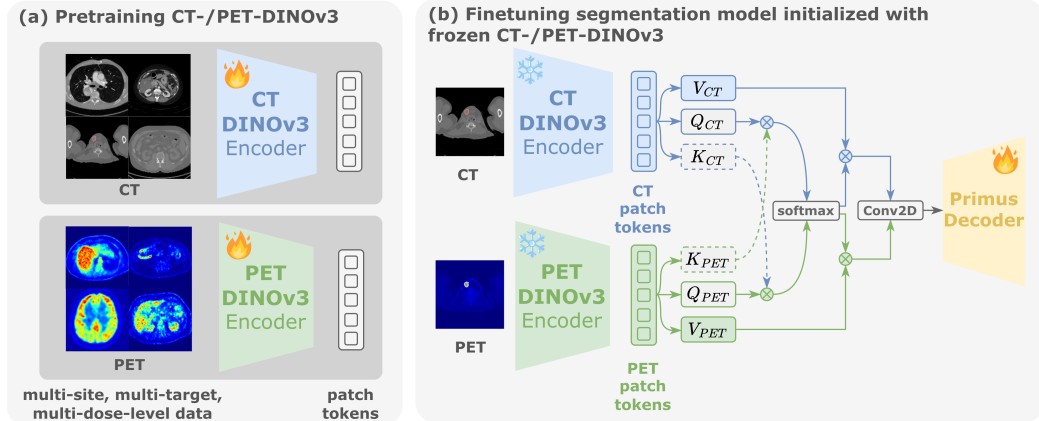

Figure 1: **Overview of the Dual-DINOv3 workflow.** (a) Domain-specific pretraining. Independent DINOv3 ViT-B/16 encoders are pretrained separately on large-scale CT and PET datasets using the three-stage self-distillation pipeline, enabling each encoder to learn modality-specialized visual representations without requiring paired PET/CT data. (b) Domain-fusion finetuning. For downstream PET/CT tumor segmentation, frozen CT-DINOv3 and PET-DINOv3 encoders process each modality independently to produce multiscale patch tokens. At each scale, the corresponding PET and CT tokens are fused through a cross-attention module enabling bidirectional exchange of anatomical (CT) and functional (PET) features. The fused multimodal features are then decoded by the Primus Multiscale decoder to generate final segmentation masks.

### 2.1. Dataset

#### 2.1.1. Pretraining Dataset

**CT Dataset** The CT dataset comprised 3.87 million 2D axial slices extracted from publicly available 3D volumes spanning multiple anatomical regions, including head-and-neck, hepatic, pulmonary, pancreatic, splenic, and colonic systems. Each volume was resampled

to 0.45 mm isotropic resolution, converted to axial slices, and spatially standardized by cropping to $256 \times 256$ pixels. Slice intensities were then normalized to the range $[0, 1]$ using min–max scaling, followed by per-slice z-score standardization to account for scanner- and protocol-dependent variations. Additional implementation details of the CT preprocessing pipeline can be found in (Li et al., 2025).

**PET Dataset** The PET dataset consisted of total-body FDG-PET scans from the ultra-low-dose PET imaging challenge dataset (UDPET) (Xue et al., 2025). This included 1,447 subjects with diverse neurological and oncological conditions and a wide range of dose reduction factors (4, 10, 20, 50, 100). Data were acquired using Siemens Biograph Vision Quadra (n = 433) and United Imaging uExplorer (n = 1,060) PET/CT systems. After preprocessing, the final PET pretraining set contained 399,684 axial slices covering the anatomical extent from brain to pelvis.

PET preprocessing required additional steps to ensure that only informative slices, particularly those containing focal radiotracer uptake, were included. A slice-wise foreground filtering procedure was applied by estimating a background intensity baseline from the lowest percentile of voxel intensities in each volume. A slice was retained only if its dilated foreground mask occupied more than 20% of the image area, thereby removing near-empty slices or slices consisting of limbs. For retained slices, the center of mass of the foreground was computed to re-center the field of view, followed by symmetric padding and cropping to a fixed $256 \times 256$ layout. Finally, PET intensities were normalized to the range $[0, 1]$ using volume level min–max scaling, where the minimum and maximum values were computed over the entire 3D scan rather than per slice.

Finally, intensities were normalized to the range $[0, 1]$ via min–max scaling. Empirically, we found this data-cleaning and foreground-filtering process to be essential for stable and successful PET-specific DINOv3 pretraining, as models trained on unfiltered slices exhibited degraded convergence and poorer representation quality.

### 2.2. Finetuning Dataset

To evaluate the impact of domain-specialized pretraining on downstream segmentation performance, we focused on the publicly available HECKTOR dataset (Oreiller et al., 2022), which provides paired PET/CT imaging with two clinically relevant target structures. The dataset contains head-and-neck PET/CT scans from 245 patients with oropharyngeal cancer treated with radiotherapy, each with expert-annotated primary gross tumor volume (GTVp; mean volume $12.71 \pm 17.28$ cm$^3$) and metastatic lymph nodes (GTVn; mean volume $13.02 \pm 18.75$ cm$^3$). PET images were acquired using integrated PET/CT systems centered on the head-and-neck region, accompanied by low-dose CT for attenuation correction and anatomical localization.

All PET/CT volumes were preprocessed uniformly. The PET and CT images were resampled in-plane to the cohort-specific median spatial resolution using cubic interpolation for CT and nearest-neighbor interpolation for segmentation masks. Axial slices were extracted and standardized to $512 \times 512$ pixels. PET intensities were normalized to $[0, 1]$ using volume level min–max scaling, whereas CT intensities were standardized using z-score normalization.

## 2.3. Domain-specific Pretraining with DINOv3

### 2.3.1. PRETRAINING OBJECTIVES

The pretraining pipeline was designed as a three-stage procedure to progressively refine semantic, structural, and multi-scale representations before downstream adaptation. **Stage 1** follows the standard self-supervised framework and jointly minimizes the global–local self-distillation loss, $L_{\text{DINO}}$, the masked image modeling loss, $L_{\text{iBOT}}$ (Zhou et al., 2022), and the feature-uniformity regularizer, $L_{\text{KoLeo}}$ (Sablayrolles et al., 2018), forming the overall objective

$$L_{\text{Stage 1}} = L_{\text{DINO}} + L_{\text{iBOT}} + 0.1\, L_{\text{KoLeo}}, \tag{1}$$

**Stage 2** augments this objective by incorporating a Gram-matrix structural consistency loss, $L_{\text{Gram}}$, which aligns second-order patch co-activation patterns between teacher and student representations. The combined objective becomes

$$L_{\text{Stage 2}} = L_{\text{DINO}} + L_{\text{iBOT}} + L_{\text{KoLeo}} + 2\, L_{\text{Gram}}, \tag{2}$$

**Stage 3** retains the losses (2), used in Stage 2 but applies them to a mixture of low- and high-resolution global and local crops, enabling the encoder to learn stable multi-scale representations suited for higher-resolution downstream PET/CT segmentation.

**Mathematical Formulation**   Let $f_s$ and $f_t$ represent the student and teacher networks, with $f_t$ parameters updated via exponential moving average (EMA) of $f_s$. For each input image $x$, we sample two global views $\{x^{(g_1)}, x^{(g_2)}\}$ and $K$ local views $\{x^{(l_1)}, \ldots, x^{(l_K)}\}$. For each view, the networks produce class-token representations:

$$
\begin{aligned}
z_s^{(g_i)} &= f_s^{\text{cls}}(x^{(g_i)}), \\
z_s^{(l_j)} &= f_s^{\text{cls}}(x^{(l_j)}), \\
z_t^{(g_i)} &= f_t^{\text{cls}}(x^{(g_i)}).
\end{aligned}
\tag{3}
$$

For masked image modeling, the student network further produces patch-token outputs

$$h_s(u) = f_s^{\text{patch}}(x)_u, \qquad u \in \mathcal{M}, \tag{4}$$

where $\mathcal{M}$ denotes the set of masked patch indices.

We additionally extract feature embeddings for regularization as bellow:

$$v_s^{(g_i)} = f_s^{\text{pre}}(x^{(g_i)}), \tag{5}$$

$$F_s = f_s^{\text{patch emb}}(x), \quad F_t = f_t^{\text{patch emb}}(x). \tag{6}$$

***Global–Local Distillation.***   We employ a DINO-style distillation objective comprising global–global and local–global alignment terms. The global-to-global loss is defined as

$$L_{\text{DINO}}^{\text{global}} = \frac{1}{N_g(N_g - 1)} \sum_{i=1}^{N_g} \sum_{\substack{j=1 \\ j \neq i}}^{N_g} \text{CE}\left( \sigma(z_t^{(g_j)}),\, \sigma(z_s^{(g_i)}) \right), \tag{7}$$

and the local-to-global loss is given by

$$L_{\text{DINO}}^{\text{local}} = \frac{1}{N_l N_g} \sum_{i=1}^{N_l} \sum_{j=1}^{N_g} \text{CE}\Big(\sigma(z_t^{(g_j)}),\, \sigma(z_s^{(l_i)})\Big). \tag{8}$$

The total distillation loss is

$$L_{\text{DINO}} = \alpha_{\text{g}}\, L_{\text{DINO}}^{\text{global}} + \alpha_{\text{l}}\, L_{\text{DINO}}^{\text{local}}. \tag{9}$$

***Masked Image Modeling.*** Following the iBOT framework, we impose consistency over masked patch tokens:

$$L_{\text{iBOT}} = \frac{1}{|\mathcal{M}|} \sum_{u \in \mathcal{M}} \text{CE}(\sigma(h_t(u)),\, \sigma(h_s(u))). \tag{10}$$

***Feature Uniformity Regularization.*** To encourage uniform coverage of the embedding space, we adopt a KoLeo-style regularization term:

$$L_{\text{KoLeo}} = -\frac{1}{N_g} \sum_{i=1}^{N_g} \log\left(\min_{j \neq i} \|v_s^{(g_i)} - v_s^{(g_j)}\|_2\right). \tag{11}$$

***Structural Consistency via Gram Matrix Alignment.*** Finally, structural alignment between teacher and student is enforced through matching their patch-level Gram matrices:

$$L_{\text{Gram}} = \frac{1}{P^2} \left\| F_s F_s^\top - F_t F_t^\top \right\|_F^2, \tag{12}$$

where $P$ denotes the number of patch tokens.

### 2.3.2. Implementation Details

We adopt the DINOv3 self-distillation framework, where the teacher network is updated as an EMA of the student parameters. The EMA decay coefficient starts at 0.996 and is linearly increased to 1.0 over the course of each training stage. Domain-specific pretraining for both CT and PET modalities is initialized from the publicly available DINOv3 ViT-B checkpoint trained on the LVD-1689M dataset (Siméoni et al., 2025).

**Stage 1.** Both student and teacher networks are trained for 100,000 iterations using the AdamW optimizer with a base learning rate of $2\times10^{-4}$, weight decay of 0.04, and momentum parameters $(\beta_1, \beta_2) = (0.9, 0.98)$. A cosine learning rate schedule is used together with a 10,000-iteration linear warmup. Training is performed with a total batch size of 1024 across two NVIDIA A6000 Pro GPUs.

**Stage 2.** A secondary Gram teacher is initialized from the EMA teacher checkpoint at 20,000 iterations of Stage 1. The Gram teacher and student encoders are then jointly optimized for an additional 10,000 iterations using a reduced learning rate of $5 \times 10^{-5}$, weight decay of 0.04, and the same AdamW momentum settings.

**Stage 3.** Both teacher and student networks are initialized from the final Stage 2 checkpoints and further optimized for 10,000 iterations using a learning rate of $2.5 \times 10^{-5}$ with cosine decay.

### 2.4. Finetuning on Downstream Tasks

The Dual-DINOv3 segmentation model is constructed using two independently pretrained DINOv3 ViT-B/16 encoders, one optimized on CT data and the other on PET data. During downstream inference and finetuning, CT images are fed exclusively into the CT-DINOv3 encoder and PET images exclusively into the PET-DINOv3 encoder, ensuring each modality is processed by its dedicated representation space. Each encoder consists of 12 Transformer blocks, and multiscale feature tokens are extracted at layers 2, 5, 8, and 11, capturing a progression from high-resolution structural detail to deep semantic context.

At each scale, the corresponding PET and CT tokens enter a cross-attention feature fusion module, adapted from the multimodal attention mechanism described in (Lin et al., 2025). The module computes trainable query, key, and value projections for each modality, enabling PET tokens to attend to CT keys and vice versa. This bidirectional attention encourages each modality to amplify features that are corroborated by the other—e.g., PET functional hotspots aligned with CT anatomical boundaries—resulting in fused representations that integrate complementary metabolic and structural cues. Cross-attended features from both streams are subsequently channel-wise combined and projected through a $1 \times 1$ convolution to produce a unified multimodal feature map at each resolution. This strategy decouples representation learning from multimodal data availability where the PET-DINOv3 and CT-DINOv3 encoders can be pretrained independently on large, modality-specific datasets, while the cross-attention module later learns to selectively pass informative PET and CT features without requiring paired PET/CT scans during pretraining.

The resulting fused multiscale representations are passed into the Primus Multiscale decoder, which reconstructs dense spatial feature maps using hierarchical upsampling and skip-aligned transformations. This decoder integrates global semantic information from deeper layers with local detail from shallower layers to produce high-resolution segmentation masks. Throughout this process, both PET-DINOv3 and CT-DINOv3 encoders remain frozen, enabling efficient training while preserving the integrity of their modality-specific pretrained representations.

This design provides a computationally efficient alternative to CNN decoders by directly inverting the ViT patch embedding process, thus is significantly more lightweight than U-Net–style CNN decoders. Each model was trained for 1,000 epochs using the AdamW optimizer with $(\beta_1, \beta_2) = (0.9, 0.98)$, a base learning rate of $3 \times 10^{-4}$, weight decay of $5 \times 10^{-2}$, cosine learning rate decay, and linear warmup during the first 50 epochs. Batch size was set to 16. Each fold was trained on a single NVIDIA A6000 Pro GPU and required approximately one day to complete.

### 2.5. Evaluation

To quantify the contributions of domain-specific pretraining and multimodal fusion, we evaluated four configurations sharing a ViT-B/16 backbone and Primus multiscale decoder, all trained under the nnU-Net protocol. The comparison isolates: (1) the value of generic

pretraining versus training from scratch, (2) the impact of increasing domain proximity between pretraining data and PET/CT segmentation, and (3) the benefit of explicitly preserving modality-separated representations.

**Model 1: Single-stream trained from scratch.** A single-stream ViT-B/16 encoder with Primus decoder initialized from random weights, serving as a baseline without pretraining.

**Model 2: Single-stream natural-image DINOv3 initialization.** A single-stream ViT-B/16 encoder initialized from the original natural-image DINOv3 checkpoint (Siméoni et al., 2025); PET and CT slices are concatenated as a pseudo-RGB input and jointly encoded, capturing the effect of generic visual pretraining without medical domain adaptation.

**Model 3: Single-stream CT DINOv3 initialization.** A single-stream ViT-B/16 encoder initialized from CT-pretrained DINOv3 model (Li et al., 2025), with PET and CT stacked as a pseudo-RGB input, assessing the benefit of medical domain specialization under a shared encoder.

**Model 4: Single-stream PET DINOv3 initialization.** A single-stream ViT-B/16 encoder initialized from the PET-pretrained DINOv3 model, with PET and CT stacked as a pseudo-RGB input, assessing the benefit of medical domain specialization under a shared encoder.

**Model 5: Dual-stream PET/CT DINOv3 initialization (Dual-DINOv3).** The proposed dual-stream architecture with independently pretrained CT-DINOv3 and PET-DINOv3 encoders; PET and CT inputs are encoded separately, fused at multiple scales via cross-attention, and decoded by the Primus multiscale decoder to isolate the effect of modality-separated pretraining and explicit cross-modality fusion.

For quantitative assessment, the micro Dice coefficient was calculated as the unweighted mean Dice similarity across subjects whereas the macro Dice coefficient weighted each subject's Dice score by its corresponding tumor volume. Performance was reported using five-fold stratified cross-validation where data were split into 60% training, 20% validation, and 20% held-out testing sets. Paired t-tests were used to assess statistical significance between model configurations.

## 3. Results

### 3.1. Impact of Pretraining Domain Alignment and Dual-Modality Representation Learning

Across both GTVp and GTVn, the dual-stream CT/PET DINOv3 configuration achieved the highest Dice scores in all lesion-size categories, significantly outperforming all other settings ($p < 0.001$). In contrast, natural-image pretraining consistently degraded performance relative to training from scratch, most notably for small GTVn lesions where micro Dice declined from 0.318 to 0.264. These results reinforce that misaligned pretraining can introduce non-transferable biases that are detrimental to PET/CT finetuning.

With increased domain proximity to the downstream task, both CT-specific and PET-specific DINOv3 pretraining substantially improved segmentation accuracy across all tumor sizes and outperformed both the from-scratch baseline and low-domain-proximity initialization. Notably, PET-specific pretraining shows consistent gains over CT-specific pretraining for small and medium lesions, consistent with PET's sensitivity to metabolically active and

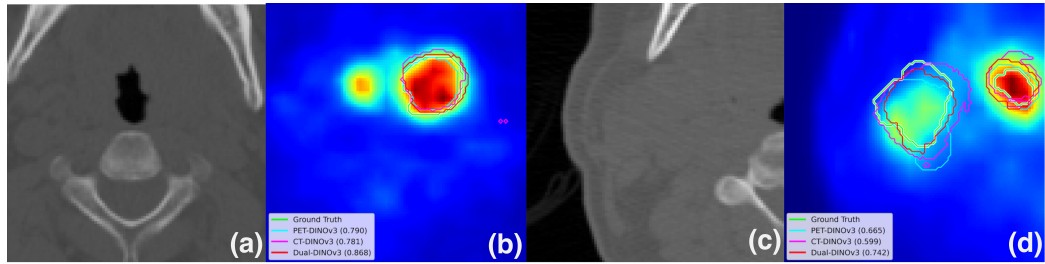

Figure 2: Segmentation performance across tumor volume subgroups. Representative cases demonstrate segmentation outcomes for (a, b) large-volume (18.91 cm3) and (c, d) small-volume (5.48 cm3) lesions. Each panel displays co-registered CT/PET images with ground-truth contours and model predictions, with the corresponding Dice score annotated.

low-contrast tumors. These improvements are less pronounced for large primary tumors in Tab1 and Fig 2. However, the current PET-specific DINOv3 encoder may not fully reflect the isolated contribution of PET representations, as it is pretrained on a sub-million-scale dataset, whereas CT-specific pretraining benefits from substantially larger data; thus, observed differences reflect both modality characteristics and pretraining scale.

Table 1: Segmentation performance stratified by tumor volume for HECKTOR dataset. Primary tumors (GTVp) and lymph nodes (GTVn) were categorized as small ($<5$ cm$^3$; n=69), medium (5–15 cm$^3$; n=102), and large ($>15$ cm$^3$; n=67).

| Model | Volume | Primary Tumor (GTVp) | | Lymph Node (GTVn) | |
|---|---|---|---|---|---|
| | | Micro Dice | Macro Dice | Micro Dice | Macro Dice |
| (1) From scratch | Small | 0.3181 (0.2175) | 0.3609 | 0.2838 (0.2697) | 0.3795 |
| | Medium | 0.5513 (0.2031) | 0.5560 | 0.6167 (0.2449) | 0.6300 |
| | Large | 0.6681 (0.1692) | 0.6506 | 0.7351 (0.1182) | 0.7234 |
| (2) Natural images pretrained | Small | 0.2636 (0.1401) | 0.2959 | 0.1249 (0.1719) | 0.1765 |
| | Medium | 0.5306 (0.1720) | 0.5354 | 0.5098 (0.1979) | 0.5329 |
| | Large | 0.6571 (0.1536) | 0.6489 | 0.6284 (0.1426) | 0.6190 |
| (3) CT pretrained | Small | 0.4629 (0.2210) | 0.4865 | 0.3461 (0.2710) | 0.4182 |
| | Medium | 0.6953 (0.1813) | 0.6983 | 0.6639 (0.2131) | 0.6783 |
| | Large | 0.8184 (0.0767) | 0.8194 | 0.7624 (0.1036) | 0.7564 |
| (4) PET pretrained | Small | 0.4718 (0.2156) | 0.5047 | 0.3674 (0.2530) | 0.4539 |
| | Medium | 0.7152 (0.1964) | 0.7178 | 0.6834 (0.2013) | 0.6991 |
| | Large | 0.8072 (0.0823) | 0.8021 | 0.7713 (0.1128) | 0.7636 |
| **(5) CT/PET pretrained** | Small | **0.5346 (0.2422)** | **0.5582** | **0.4423 (0.3156)** | **0.5432** |
| | Medium | **0.7589 (0.1621)** | **0.7611** | **0.7061 (0.2089)** | **0.7185** |
| | Large | **0.8544 (0.0637)** | **0.8584** | **0.8079 (0.0789)** | **0.8033** |

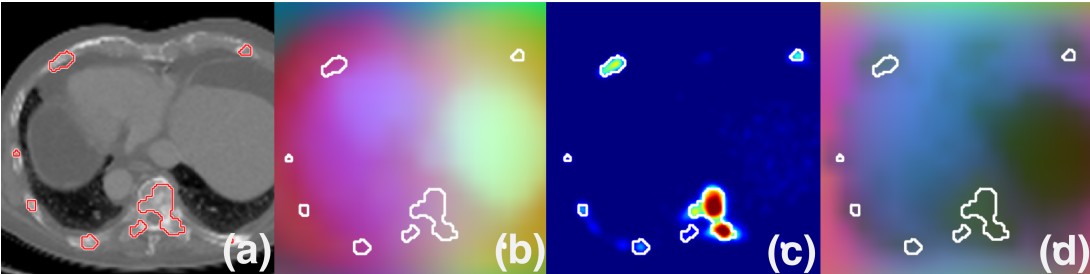

Figure 3: Principal component analysis of dense feature representations derived from modality-specific DINOv3 encoders. (a, b) CT slice and its corresponding PCA map generated from CT-DINOv3 patch tokens. (c, d) PET slice and its corresponding PCA map generated from PET-DINOv3 patch tokens. Tumor contours are shown in red on the input images and in white on the PCA visualizations.

### 3.2. Principal Component Analysis Visualization of DINOv3 Dense Feature Maps

To interpret the learned representations, we performed principal component analysis (PCA) on the dense patch-token embeddings produced by the modality-specific DINOv3 models. The first three principal components were mapped to RGB space for visualization, providing qualitative insight into how CT-DINOv3 and PET-DINOv3 attend to anatomical and functional structures, respectively.

For CT inputs (Figure 3a), PCA visualizations (Figure 3b) showed strong correspondence with anatomical boundaries, including soft-tissue interfaces, air–tissue transitions, and bony structures. These activations indicate that the CT-pretrained encoder successfully learned structural priors characteristic of CT imaging, supporting its effectiveness in tasks requiring precise spatial localization.

In contrast, PCA maps derived from PET inputs revealed activation patterns concentrated around physiologic tracer uptake regions (Figure 3c). Importantly, even small focal lesions were represented by distinct color clusters in the corresponding PCA projection (Figure 3d), demonstrating that PET-DINOv3 is sensitive to functional information. Although the model does not fully disentangle malignant from normal physiologic uptake, the clear separation of low-uptake background from focal abnormalities suggests that PET-DINOv3 contributes complementary functional cues that are absent in CT features alone. This aligns with the empirical performance gains observed when transitioning from the single-stream CT-DINOv3 model (Model 3) to the dual-stream PET/CT configuration (Model 5).

These observations highlight the potential for further enhancing PET-specific representations. Future work could incorporate anatomical priors directly into PET pretraining—for example through 2.5D context aggregation or extend frozen 2D encoders into 3D architectures via weight inflation to better capture volumetric metabolic patterns during finetuning.

## 4. Discussion

**Comparison with Existing PET/CT Segmentation and Foundation Models**   This work examines how domain-aligned self-supervised pretraining and modality-separated representation learning influence PET/CT tumor segmentation under realistic public-data constraints. Rather than pursuing exhaustive external benchmarking, our evaluation emphasizes controlled ablations designed to isolate the effects of pretraining domain alignment and multimodal fusion, allowing us to focus on representation learning behavior while situating the proposed approach within the broader PET/CT segmentation literature. Several fully supervised approaches evaluated in the HECKTOR challenge have reported strong segmentation performance, with top entries achieving aggregated Dice scores of approximately 0.78–0.79 in the HECKTOR 2022 overview (Oreiller et al., 2022). These methods typically rely on end-to-end supervised pipelines optimized specifically for the benchmark, often incorporating ensembling, multi-stage localization, and extensive task-specific tuning. In contrast, the objective of this work is not to maximize HECKTOR performance through task-specific supervised engineering, but to establish a publicly reproducible PET-specific foundation encoder and to study how modality-aligned pretraining and modality-separated fusion affect downstream PET/CT learning. Within this setting, Dual-DINOv3 achieves competitive performance while relying on frozen, publicly pretrained encoders and minimal task-specific optimization.

The most closely related PET/CT tumor segmentation foundation model is the work of (Oh et al., 2025). However, no official implementation or pretrained weights are publicly released, and the reported results are obtained using private datasets, preventing fair and reproducible evaluation under the HECKTOR protocol. Other PET foundation models, such as SegAnyPET (Zhang et al., 2025), as well as general medical foundation models such as MedSAM (Ma et al., 2024), adopt prompt-based segmentation paradigms. These methods primarily target organ or general-purpose segmentation and require spatial prompts to guide inference. In contrast, PET/CT tumor segmentation involves lesions that are morphologically diverse, heterogeneous in metabolic uptake, and often small or poorly defined. As a result, segmentation performance in this setting is highly sensitive to prompt type, placement, and quantity. Prior studies have shown that SAM-based methods exhibit substantial performance variability under different prompting strategies (Mazurowski et al., 2023; Cheng et al., 2023; Wang et al., 2024), making it difficult to define a standardized and fair prompting protocol for comparison with prompt-free automatic tumor segmentation. For these reasons, prompt-based approaches are considered complementary rather than included as direct empirical baselines. Taken together, differences in supervision level, task formulation, data availability, and reproducibility constraints motivate our focus on controlled ablations rather than direct numerical comparison with these methods in a single experimental table.

**Choice of Self-Supervised Pretraining Strategy**   Recent benchmarks report that DINOv3-style self-distillation yields highly transferable representations and outperforms reconstruction-based approaches such as MAE on medical image segmentation tasks (Liu et al., 2025; Li et al., 2024), motivating our choice of DINOv3 for PET representation learning. While natural-image–pretrained MAE checkpoints could be evaluated on PET/CT segmentation, MAE models typically reach their full potential only with domain-aligned

pretraining and downstream adaptation (He et al., 2021; Tang et al., 2021; Huang et al., 2024). As such, results from natural-image MAE would largely reflect domain mismatch rather than intrinsic differences in SSL objectives. A fair comparison would require MAE to be pretrained directly on large-scale PET or PET/CT datasets under matched protocols, representing a distinct large-scale benchmarking effort that we identify as an important direction for future work.

**Implications and Future Directions** In summary, a key contribution of this study is the release of a fully public PET-specific DINOv3 encoder pretrained at scale on FDG-PET data. While PET/CT tumor segmentation serves as a representative validation task, the broader value of a PET foundation model lies in its versatility. The proposed PET-DINOv3 encoder can be applied to a wide range of PET applications beyond segmentation, including classification, representation learning, registration, and image synthesis. Future work may explore integration with task-specific pipelines, extension to 2.5D or 3D modeling, and comprehensive cross-SSL benchmarking as PET-scale pretraining of alternative objectives becomes feasible.

## 5. Conclusion

This work introduces Dual-DINOv3, a dual-stream, modality-separated DINOv3 framework for PET/CT representation learning and downstream tumor segmentation. Our first contribution is the release of a fully public PET-DINOv3 encoder, pretrained using the three-stage DINOv3 self-distillation pipeline on large-scale FDG-PET datasets. This encoder provides a general-purpose PET representation that can be readily incorporated into dense prediction tasks such as segmentation, registration, and image synthesis.

We further demonstrate that domain alignment between pretraining and downstream tasks has a substantial impact on PET/CT segmentation performance. Natural-image initialization degrades performance relative to supervised training from scratch, whereas domain-specific DINOv3 pretraining consistently improves accuracy. Motivated by the practical challenge of acquiring large paired PET/CT datasets, our second contribution is a dual-stream architecture that decouples modality-specific pretraining from multimodal finetuning. By independently pretraining PET- and CT-DINOv3 encoders and fusing their representations through cross-attention, Dual-DINOv3 leverages complementary functional and anatomical cues without requiring paired data during pretraining.

## Acknowledgments

This research is supported in part by the National Institutes of Health, United States under Award Numbers R01EB032680 and R01CA272991.

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
