# OpenReview forum: "Beyond Natural Images: A Dual-Stream DINOv3 Framework for PET/CT Segmentation"
_MIDL.io/2026/Conference — MIDL 2026 Poster_

### Official Review · Reviewer_aGMv · 2026-01-06

**Confidence:** 4
**Preliminary Rating:** 4

**Summary:**

This paper introduces a dual-stream framework for PET/CT tumor segmentation. It addresses two key limitations: the lack of a public, PET-specific pre-trained encoder and the reliance on fully paired PET/CT data for multimodal training. Evaluated on the HECKTOR benchmark, the model significantly outperformed baselines, demonstrating that domain-aligned pre-training and explicit multimodal fusion are crucial for optimal performance, especially compared to models pre-trained on misaligned natural images.

**Strengths:**

+ Medical image segmentation aligns well with the theme of MIDL.
+ The application of DINOv3 effectively reflect the latest advancements in the field of medical image segmentation.
+ The paper is generally well-written and easy to follow.
+ Extensive experiments on the large-scale dataset demonstrate the superiority of the proposed PET/CT-DINOv3.

**Weaknesses:**

- It is recommended to include the complete structure of the Primus Decoder in Figure 1 to facilitate reader comprehension.
- It is suggested to add comparisons, or at least, discussions, with SAM-based medical image segmentation methods, including [*1,*2], to justify the use of DINO for pre-training.

[*1] Samed-2: Selective memory enhanced medical segment anything model
[*2] Sam2-unet: Segment anything 2 makes strong encoder for natural and medical image segmentation

**Detailed Comments:**

See weaknesses.

**Justification Of The Preliminary Rating:**

Good novelty, comprehensive experiments. See strengths:
+ Medical image segmentation aligns well with the theme of MIDL.
+ The application of DINOv3 effectively reflect the latest advancements in the field of medical image segmentation.
+ The paper is generally well-written and easy to follow.
+ Extensive experiments on the large-scale dataset demonstrate the superiority of the proposed PET/CT-DINOv3.

**Questions To Address In The Rebuttal:**

See weaknesses.

---

> ### Author Response · Authors · 2026-01-24
>
> We sincerely thank Reviewer aGMv for the positive evaluation of our work and for recognizing the novelty and relevance of introducing a public PET-specific DINOv3 encoder and a modality-separated PET/CT framework. We are encouraged by the reviewer’s assessment that the paper is well written and that the experimental results demonstrate the effectiveness of domain-aligned pretraining and multimodal fusion. Below, we address the remaining comments point by point.
>
> 1. Include complete Primus Decoder structure in Fig.1.
>
> We thank the reviewer for the suggestion. We note that the Primus decoder is not originally proposed in this work but adopted from prior literature. To avoid redundancy and potential inconsistency with the original design, we chose to reference the detailed architecture description in the original Primus publication rather than reproducing its full internal structure in Figure~1. We have clarified this in the manuscript and added an explicit citation directing readers to the original work for comprehensive architectural details.
>
> 2. Add comparisons or discussion with SAM-based medical image segmentation methods (e.g., Samed-2, Sam2-UNet)
>
> We appreciate the suggestion to discuss SAM-based segmentation models to further justify the choice of DINOv3 for pretraining.
>
> In line with comments from other reviewers, we added an expanded discussion in the Discussion section comparing our approach with SAM-based medical segmentation methods, including MedSAM, Samed-2, and Sam2-UNet. We clarify that these models adopt a prompt-based segmentation paradigm, whereas our work focuses on prompt-free automatic tumor segmentation. Prior studies have shown that SAM-based performance is highly sensitive to prompt type, prompt placement, and prompt quantity, making it difficult to define a standardized and fair prompting protocol for comparison with prompt-free settings.
>
> We further note that many SAM-based methods are primarily designed for organ or general-purpose segmentation, where target structures are large, spatially coherent, and anatomically constrained. In contrast, PET/CT tumor segmentation involves lesions that are morphologically diverse, heterogeneous in metabolic uptake, and often small or poorly defined. As a result, segmentation performance in this setting is highly sensitive to the choice, number, and placement of prompts, making few-shot or prompt-driven paradigms difficult to standardize and evaluate fairly. Consequently, SAM-based approaches are better viewed as complementary tools rather than direct baselines for prompt-free automatic tumor segmentation. These distinctions are now explicitly discussed in the manuscript to justify our choice of DINOv3 and to clarify the scope of comparability.

---

> ### Comment · Area_Chair_YGzM · 2026-02-01
> **For Reviewer - Please update your final rating after reviewing the author's response.**
>
> Hello there, please update your final rating after reviewing the author's response. Thank you for your time and support.

---

### Official Review · Reviewer_kRjX · 2026-01-09

**Confidence:** 4
**Preliminary Rating:** 2
**Final Rating:** 2

**Summary:**

This study begins from the DINOv3 framework which is separately trained for PET and CT images. These pre-trained encoders are then combined with a multimodal cross-attention decoder to output the segmentation mask. All the models are trained on 2D slices from the initial images to segment tumor regions. Interpretation via PCA visualization is conducted along with an ablation study to support the claims and participation of each component. Its significance would be clearer with stronger benchmarking against established PET/CT tumor segmentation and self-supervised pretraining baselines.

**Strengths:**

1) Using separate encoders for PET and CT is a sensible design choice for pretraining, particularly when one modality is not provided.
2) Training and evaluation on a public dataset improves reproducibility and comparability, although releasing code would further strengthen this aspect
3) The inclusion of ablation experiments and qualitative/interpretability analyses (e.g., PCA visualizations) helps clarify the role of each module and improves transparency.

**Weaknesses:**

1) The method primarily combines two previously introduced components, DINOv3-style encoders and a cross-attention decoder, largely in a standard configuration. Without clearer methodological innovations (e.g., new objectives, fusion strategies, or training procedures), the contribution reads mainly as an effective integration.
2) The paper does not provide an adequate benchmark against recent state-of-the-art methods for PET/CT tumor segmentation and related multimodal pretraining strategies. The main quantitative table appears focused on internal ablations rather than external baselines, which makes it difficult to assess competitiveness.

**Detailed Comments:**

A figure with segmentation results for visual inspection would enhance the results' presentation.

**Justification Of Final Rating:**

I thank the authors for their detailed rebuttal and for adding a discussion regarding state-of-the-art methods.
However, I maintain my recommendation of Weak Reject. My initial concerns regarding the paper's contribution and evaluation remain largely unresolved:
- Limited Novelty: The method remains primarily an integration of existing components (DINOv3 encoders and cross-attention decoders) without significant methodological innovations in objectives or fusion strategies.
- Missing Quantitative Benchmarks: While the authors added discussion regarding SOTA methods, the manuscript still lacks a direct quantitative evaluation comparing their approach to other SSL pretraining strategies. Such a comparison will have provided evidence on the selection of DINO as a pretraining strategy and whether other techniques are more suitable for PET images.

**Justification Of The Preliminary Rating:**

Based on the current evidence, I would lean toward weak reject. The approach is well-motivated and the modality-specific pretraining plus fusion design appears promising, but the manuscript does not yet demonstrate clear algorithmic novelty or provide sufficiently strong comparisons to state-of-the-art segmentation and pretraining baselines. If the authors can clarify the unique methodological contributions and add robust external benchmarking (with standard metrics and consistent protocols), the overall assessment could improve substantially.

**Questions To Address In The Rebuttal:**

1)What are the key new methodological contributions beyond combining a DINOv3-based pretraining setup with a cross-attention decoder? Please specify what is novel in the objectives, fusion mechanism, architecture, or training protocol
2) What are the reported segmentation results (e.g., DSC) of recent peer-reviewed PET/CT tumor segmentation methods on the same dataset (or a clearly comparable setting)? Additional experiments would support the superiority of the algorithm. The same applies to other self-supervised pretraining strategies, e.g. MAE (Masked Autoencoders), SwinMM, VICReg.

---

> ### Author Response · Authors · 2026-01-24
>
> We sincerely thank Reviewer kRjX for the thoughtful review and constructive suggestions. We appreciate the recognition of the motivation for modality-separated PET/CT pretraining, the use of public datasets, and the inclusion of ablation and interpretability analyses. Below, we address each concern in detail and summarize the corresponding revisions to the manuscript.
>
> 1. Mainly integration; unclear what is new beyond combination.
>
> We agree that cross-attention itself is a standard fusion mechanism. Accordingly, we revised the manuscript to clarify that the primary contribution of this work does not lie in proposing a new attention operator in isolation, but in how representation learning is structured under realistic PET data constraints. In particular, publicly available PET datasets are substantially smaller than those available for other medical modalities and natural images, and large-scale paired PET/CT datasets suitable for foundation-model pretraining remain scarce.
>
> In this context, a key contribution of the present work is demonstrating that PET-specific DINOv3 pretraining is feasible and effective at a sub-million 2D slice scale, which is orders of magnitude smaller than the billion-scale natural-image data used for DINOv3 and even substantially smaller than the multi-million-image medical corpora used in MedDINOv3. This result provides practical evidence that domain-aligned PET foundation models can be learned under realistic public-data availability, rather than assuming access to extremely large or private datasets.
>
> Building on this insight, we introduce a modality-separated foundation modeling strategy that pretrains PET and CT encoders independently on unpaired data and integrates them only during downstream finetuning. This design directly addresses the paired-data bottleneck in multimodal PET/CT learning and enables scalable, reproducible foundation models under realistic conditions. Moreover, the resulting PET-DINOv3 encoder is not task-specific and can be reused for applications beyond segmentation, including classification, registration, and image synthesis.
> To sharpen this novelty framing, we revised the Introduction to clearly distinguish what is new (the public PET-DINOv3 encoder, modality-separated pretraining strategy, and systematic evidence of domain-misalignment effects); and what is standard (cross-attention fusion).
>
> 2. Benchmarking against state-of-the-art PET/CT segmentation and SSL baselines.
>
> We thank the reviewer for emphasizing the importance of broader benchmarking, which was also raised by another reviewer. In response, we added a dedicated Discussion section to contextualize our experiments with respect to both state-of-the-art PET/CT segmentation methods and alternative SSL pretraining strategies.
>
> Regarding SSL baselines, we expanded the discussion to clarify our choice of DINOv3. Recent benchmark studies have shown that DINOv3-style self-distillation yields highly transferable representations and consistently outperforms reconstruction-based approaches such as MAE pretrained on natural images for medical image segmentation tasks (e.g., Liu et al., 2025). These findings motivate our use of DINOv3 as the backbone for PET representation learning.
>
> In principle, publicly released natural-image-pretrained MAE checkpoints could be evaluated on downstream PET/CT tumor segmentation. However, MAE models are known to exhibit their strongest performance only after appropriate domain-aligned pretraining and downstream adaptation (He et al., 2022; Tang et al., 2022). As such, comparisons based solely on natural-image-pretrained MAE models would primarily reflect domain mismatch rather than differences in SSL objectives. A more informative and fair comparison would require PET- or PET/CT-specific MAE pretraining under matched protocols, which constitutes a separate large-scale benchmarking study rather than a minor additional baseline experiment.
>
> We therefore focus our empirical evaluation on isolating the effect of modality-aligned DINOv3 pretraining under public-data constraints, while explicitly positioning comprehensive cross-SSL benchmarking (including PET-specific MAE, SwinMM, and VICReg) as an important direction for future work.
>
> 3. Add segmentation visual figure
>
> We thank the reviewer for this suggestion, which was also raised by another reviewer. We agree that qualitative visualizations improve interpretability and provide complementary evidence to the quantitative results. In response, we added a new figure in the Results section showing representative PET/CT segmentation overlays.

---

> ### Comment · Area_Chair_YGzM · 2026-02-01
> **For Reviewer - Please update your final rating after reviewing the author's response.**
>
> Hello there, please update your final rating after reviewing the author's response. Thank you for your time and support.

---

### Official Review · Reviewer_Jw91 · 2026-01-12

**Confidence:** 4
**Preliminary Rating:** 2
**Final Rating:** 3

**Summary:**

The authors propose Dual-DINO, a dual-stream framework for PET/CT segmentation that aims to address two limitations in existing multimodal learning approaches: (1) the lack of a publicly available PET-specific pretrained encoder, and (2) the reliance on fully paired PET/CT data during multimodal pretraining. To this end, the authors introduce the first PET-specific DINOv3 encoder and a modality-separated framework in which PET- and CT-specific encoders are pretrained independently using modality-specific datasets. The encoders are subsequently fused during fine-tuning via multiscale cross-attention. Experiments on the public HECKTOR tumor segmentation benchmark demonstrate that the proposed approach achieves the best overall performance among the evaluated methods.

**Strengths:**

Unlike prior work that requires paired PET/CT data for multimodal pretraining, the proposed method pretrains PET and CT encoders independently using modality-specific public datasets, relying on paired data only during fine-tuning. This significantly reduces data constraints and is a meaningful contribution to the PET imaging community.

The introduction of a PET-specific pretrained encoder based on DINOv3 fills an important gap in current medical imaging resources.

On the HECKTOR benchmark, the proposed method outperforms all compared baselines, demonstrating strong empirical effectiveness.

**Weaknesses:**

The technical novelty is not very significant for the paper since the contribution is mainly utilizing the DINOv3 backbone for modality-specific pre-training, and the cross-attention is also a commonly used technique to fuse two modalities.

The experimental comparisons are restricted to different initialization or pretraining strategies (training from scratch, natural image initialization, and CT initialization). A more comprehensive evaluation should include comparisons with recent medical foundation models, such as MedSAM, SegAnyPET, and the foundation model proposed by Oh et al., 2025.

Qualitative visualizations of segmentation results are missing; only quantitative metrics are reported. Visual comparisons would improve interpretability and support the reported gains.

The method operates on 2D slices rather than 3D volumes, which may introduce inter-slice inconsistency. This limitation is not sufficiently discussed or analyzed.

**Detailed Comments:**

The PCA visualizations suggest that PET features capture meaningful structure but CT features appear less informative. If CT-only representations are weak, the observed gains from the dual-stream setup may primarily stem from the PET branch. In this case, a comparison against a single PET-stream model would be more appropriate and informative than a comparison against a single CT stream.

PET intensities are normalized to the range [0,1]. However, if this normalization is performed per slice, the same absolute uptake value may correspond to different normalized values across slices depending on the slice-wise maximum. This choice may affect feature consistency and should be clearly clarified and justified.

For experiments using natural image initialization or CT-only initialization, the authors state that PET and CT images are concatenated to mimic RGB channels. However, concatenating PET and CT produces only two channels. Please clarify how this setup aligns with RGB-based initialization and justify whether this concatenation is appropriate.

**Justification Of Final Rating:**

I would like to thank the authors for providing additional information and further clarifications. Since the contribution is leaning more toward practical application to PET, but technically more like leveraging existing methods, I am increasing my rating to borderline.

**Justification Of The Preliminary Rating:**

While the paper provides a useful domain-specific contribution and achieves strong empirical performance, the overall technical novelty is limited. Additionally, several important evaluations and analysis are missing. These factors limit the impact of the work despite its practical relevance.

**Questions To Address In The Rebuttal:**

See weaknesses and comments

---

> ### Author Response · Authors · 2026-01-24
>
> We sincerely thank Reviewer Jw91 for the detailed and constructive feedback. We appreciate the recognition of (i) our modality-separated pretraining that reduces reliance on paired PET/CT data and (ii) the value of releasing a PET-specific pretrained encoder. Below we respond point by point and summarize the corresponding revisions.
>
> 1. Limited technical novelty.
>
> We clarify that our primary contribution is not a novel fusion operator, but the establishment of a fully public PET-specific DINOv3 encoder and a paired-data-agnostic multimodal pretraining strategy that decouples modality-specific representation learning from multimodal supervision. This directly addresses two practical bottlenecks in PET foundation model research: (i) the lack of a public, reproducible PET pretrained encoder, and (ii) the difficulty of assembling large and paired PET/CT corpora for pretraining.
>
> To sharpen the novelty framing, we revised the Introduction that clearly shows what is new (public PET-DINOv3 encoder; modality-separated pretraining) from standard components (cross-attention fusion).
>
> 2. Missing comparisons to recent foundation models and other SSL strategies.
>
> We thank the reviewer for highlighting the importance of external benchmarking. In response, we added a dedicated Discussion section that situates our method with respect to recent PET/CT foundation and segmentation models and clarifies the scope and assumptions of comparative evaluations.
>
> The most closely related work is Oh et al. (2025); however, no official implementation or pretrained weights are publicly released, and their reported results rely on private datasets, preventing fair and reproducible evaluation under the HECKTOR protocol. Other PET foundation models, such as SegAnyPET (Zhang et al., 2025), and general medical models such as MedSAM (Ma et al., 2024), adopt prompt-based segmentation paradigms and primarily target organ segmentation rather than automatic tumor segmentation. Prior studies have demonstrated that SAM-based performance is highly sensitive to prompt design and prompt type (Mazurowski et al., 2023; Cheng et al., 2023; Wang et al., 2024), making it difficult to define a standardized prompt protocol for fair comparison with our prompt-free setting.
>
> Regarding alternative SSL pretraining strategies, existing benchmark studies report that DINOv3 yields more transferable representations than MAE-based approaches for medical image segmentation tasks (Liu et al., 2025; Li et al., 2024). While natural-image-pretrained MAE checkpoints could in principle be evaluated, MAE models typically demonstrate their strongest performance only after domain-aligned pretraining and downstream adaptation. A meaningful comparison would therefore require PET- or PET/CT-specific MAE pretraining, which is beyond the scope of this rebuttal due to substantial training cost.
>
> To contextualize competitiveness, we note that fully supervised HECKTOR challenge submissions report aggregated Dice scores of approximately 0.78-0.79 (Oreiller et al., 2022). Our results fall within this performance regime, supporting the effectiveness of the proposed framework.
>
>
> 3. Missing qualitative visualizations.
>
> We add a new Results figure showing representative PET/CT segmentation overlays.
>
> 4. Limitation on 2D slice-based training.
>
> We acknowledge that 2D training may introduce inter-slice inconsistency. This choice is motivated by DINOv3 being pretrained on 2D images and computational practicality. Importantly, the DINOv3 paper shows that 2D-pretrained models generalize well to video segmentation and tracking without video-specific pretraining, indicating strong temporal consistency and suitability for extension to volumetric medical imaging.
>
> 5. Comparison to PET-only single stream.
>
> To test whether gains are driven solely by PET, we add a PET-only baseline (denoted as the new Model 4) to compare individual PET and CT contributions with joint PET/CT streams. Results and Discussions are updated accordingly.
>
> 6. PET intensity normalization.
>
> PET intensities are normalized per volume (scan-level) than per slice to avoid slice-wise intensity inconsistency.
>
> 7. RGB channel handling.
>
> All ViT-B/16 encoders used in our study (including natural-image DINOv3, CT-DINOv3, and PET-DINOv3) are designed to accept 3-channel inputs. For single-stream configurations (Models 1–4), where PET and CT are concatenated as a pseudo-RGB input, we construct a 3-channel tensor as follows: PET and CT occupy two channels, and the remaining channel is filled by replicating one of the modalities (PET or CT) to maintain consistency with the 3-channel input requirement. This replication ensures architectural compatibility with pretrained weights, allowing fair reuse of pretrained parameters without modifying the patch embedding layer.

---

> ### Comment · Area_Chair_YGzM · 2026-02-01
> **For Reviewer - Please update your final rating after reviewing the author's response.**
>
> Hello there, please update your final rating after reviewing the author's response. Thank you for your time and support.

---

### Author Rebuttal · Authors · 2026-01-25

**Rebuttal:**

We thank the reviewers for their detailed and constructive feedback. In response, we substantially revised the manuscript to clarify the methodological contributions, strengthen empirical evidence, and better contextualize the work within the PET/CT segmentation and self-supervised learning literature. Below we summarize how the main concerns were addressed.

1. Clarifying methodological innovation beyond integration.

Several comments questioned whether the contribution goes beyond integrating existing components. We clarify that the primary novelty is the first large-scale adaptation and pretraining of DINOv3 on PET imaging, resulting in a publicly reproducible PET-specific foundation encoder. To our knowledge, no prior work has demonstrated that DINOv3 can be successfully trained on functional PET data at scale. We further show that PET-specific DINOv3 pretraining is feasible using sub-million-scale 2D slices, providing practical evidence that PET foundation models can be learned under realistic public-data constraints and enabling applications beyond segmentation (e.g., registration and synthesis).

Building on this, we introduce a modality-separated strategy that allows PET and CT encoders to be pretrained independently on unpaired data and fused only during downstream finetuning, directly addressing the scarcity of large and public paired PET/CT datasets.

2. Empirical additions and experimental revisions.

We added (i) a PET-only single-stream baseline, (ii) qualitative segmentation visualizations comparing Dual-DINO with the strongest single-stream baseline, and (iii) clarifications to preprocessing and implementation details.

3. Contextualization and scope of benchmarking.

We added a new Discussion section comparing our results with existing PET/CT segmentation methods, PET foundation models, and prompt-based approaches. We clarify that prompt-based models rely on different task assumptions and are highly sensitive to prompt design, making standardized comparison with our prompt-free setting challenging. Requests for broader SSL benchmarking are addressed by citing recent studies showing that DINOv3 yields more transferable representations than other methods pretrained on natural images for medical segmentation. A fair comparison would require PET- or PET/CT-specific pretraining of these alternatives under matched protocols, which we consider as a separate large-scale benchmarking effort for future work.

**Supporting Material:**

/attachment/568b98f3f5973c9f20ec792a68226c5660dc3311.pdf

---

### Meta-Review · Area_Chair_YGzM · 2026-02-09

**Recommendation:** Accept (Poster)
**Confidence:** 5

**Metareview:**

This paper presents Dual-DINO, a dual-stream PET/CT segmentation framework built on modality-specific DINOv3 pretraining, addressing two practical limitations in current PET foundation modeling: the lack of a public PET-specific pretrained encoder and the dependence on paired PET/CT data during pretraining. The reviewers consistently recognized the value of the proposed modality-separated pretraining strategy, the release of a public PET-DINOv3 encoder, and the strong empirical performance on the public HECKTOR benchmark. While concerns were raised regarding limited algorithmic novelty, given the use of established components such as DINOv3 and cross-attention, the authors clearly positioned the contribution as a domain-aligned, reproducible foundation model under realistic PET data constraints rather than a novel fusion mechanism. The rebuttal satisfactorily clarified design choices, added qualitative visualizations, introduced a PET-only baseline, and expanded discussion on missing comparisons and limitations of 2D training. Overall, the work offers a meaningful and timely contribution to PET/CT representation learning with solid experimental evidence and high practical relevance, making it well-suited for presentation as a poster at MIDL.

---

### Decision · Program_Chairs · 2026-02-13

Accept (Poster)